Ultrasound-promoted synthesis of 2-organoselanyl-naphthalenes using Oxone® in aqueous medium as an oxidizing agent

Perin Gelson 1 gelson_perin@ufpel.edu.br
Araujo Daniela Rodrigues 1
Nobre Patrick Carvalho 1
http://orcid.org/0000-0001-7920-3289 Lenardao Eder João 1 elenardao@uol.com.br
Jacob Raquel Guimarães 1
Silva Marcio Santos 2
Roehrs Juliano Alex 3
1 Laboratório de Síntese Orgânica Limpa—LASOL, Centro de Ciencias Quimicas, Farmaceuticas e de Alimentos—CCQFA, Universidade Federal de Pelotas , Pelotas, Rio Grande do Sul , Brazil
2 Centro de Ciências Naturais e Humanas—CCNH, Universidade Federal do ABC , Santo André, São Paulo , Brazil
3 Instituto Federal de Educação Ciência e Tecnologia Sul-rio-grandense—IFSul , Pelotas, Rio Grande do Sul , Brazil
El-Hiti Gamal
Electronic publication date: 2018 May 7
Publication date: 2018
Volume: 6
Electronic Location ID: e4706
Received 2018 Mar 22; Accepted 2018 Apr 15
Copyright: © 2018 Perin et al.
Copyright year: 2018
Copyright holder: Perin et al.
License: This is an open access article distributed under the terms of the Creative Commons Attribution License, which permits unrestricted use, distribution, reproduction and adaptation in any medium and for any purpose provided that it is properly attributed. For attribution, the original author(s), title, publication source (PeerJ) and either DOI or URL of the article must be cited.
License URL: https://creativecommons.org/licenses/by/4.0/

Keywords: Green chemistry, Organoselenium, Ultrasound, Oxone, Naphthalenes, Organic synthesis

Funding: The Brazilian Council for Research and Technology (CNPq) CAPES FAPERGS This work was supported by The Brazilian Council for Research and Technology (CNPq), CAPES and FAPERGS. The funders had no role in study design, data collection and analysis, decision to publish, or preparation of the manuscript.

==============================
A green methodology to synthesize 2-organoselanyl-naphthalenes based on the reaction of alkynols with diaryl diselenides is described. The electrophilic species of selenium were generated in situ, by the oxidative cleavage of the Se–Se bond of diaryl diselenides by Oxone® using water as the solvent. The reactions proceeded efficiently under ultrasonic irradiation as an alternative energy source, using a range of alkynols and diorganyl diselenides as starting materials. Through this methodology, the corresponding 2-organoselanyl-naphthalenes were obtained in moderate to good yields (56–94%) and in short reaction times (0.25–2.3 h).

Introduction

Compounds containing chalcogen atoms (S, Se, Te) are versatile synthetic intermediates for the synthesis of complex molecules (Mukherjee et al., 2010; Beletskaya & Ananikov, 2011; Godoi, Schumacher & Zeni, 2011). Furthermore, the interest in organochalcogen compounds is connected to their well reported pharmacological activities (Santi, 2014), including antidepressant-like (Brod et al., 2017), antiviral (Sartori et al., 2016), antifungal (Venturini et al., 2016), anxiolytic (Reis et al., 2017), anticholinesterasic (Peglow et al., 2017), anti-inflammatory (Pinz et al., 2017), and antioxidant (Nobre et al., 2017).

The plethora of methods to incorporate organoselenium groups in organic substrates includes the use of nucleophilic (Iwaoka, 2011), radical (Nomoto et al., 2013), and electrophilic species of selenium (Santi & Tidei, 2013; Sancineto et al., 2016). The reaction of diorganyl diselenides with a halogen source is the most used method to access electrophilic selenium species (Raucher, 1977; Azeredo et al., 2014; Shi, Yu & Yan, 2015; Rafique et al., 2016; Silva et al., 2017). However, the obtained selanyl halides are unstable and difficult to prepare (Santi, 2014). Due to the disadvantages of the use of halogenated selenium species, new alternatives have been described in the literature for the generation of electrophilic selenium species, such as the use of inorganic salts such as sodium (Kibriya et al., 2017), potassium (Santi & Tidei, 2014; Prasad et al., 2013), and ammonium (Tiecco et al., 1989; Santi et al., 2008; Santoro et al., 2010) persulfate through the in situ reaction with diorganyl diselenides. Naphthalenes and their derivatives are known for their countless biological properties reported in the literature, like anticancer (Norton et al., 2008), antifungal (Iverson & Uetrecht, 2001), and antiviral activities (Yeo et al., 2005). In addition, these compounds demonstrated a wide spectrum of applications in materials (Lee, Noll & Smith, 2008) and polymer chemistry (Reddy et al., 2007). The numerous methodologies to prepare this class of compounds include chemical modifications in functionalized naphtalenes (Koz, Demic & Icli, 2016; Aksakal et al., 2017), cyclization of alkynes and aldehydes using iron (Zhu et al., 2013) or boron (Xiang et al., 2013) catalysis, reaction of internal, and terminal alkynes with enamine and hypervalent iodine (Gao, Liu & Wei, 2013), cascade reaction of aldehydes and ketones catalyzed by trifluoromethanesulfonic acid (Manojveer & Balamurugan, 2015) and Claisen rearrangement using vanillin derivatives (Chan et al., 2017).

The synthesis of selenium-containing naphthalene derivatives, however, is scarcely described. Five main synthetic routes have been developed to construct selanyl naphthalenes: (i) annulation of aryl enynes (Yang et al., 2014), (ii) metal-catalyzed direct selenylation of naphthylboronic acids (Mohan et al., 2015), (iii) cyclization reactions of 4-arylbut-3-yn-2-ols with electrophilic selenium species, like PhSeBr (Zhang, Sarkar & Larock, 2006) or PhSeSePh/FeCl3 system (Recchi, Back & Zeni, 2017), (iv) [4+2] cycloaddition reaction of chalcogenoalkynes with o-alkynylbenzaldehydes (Mantovani, Back & Zeni, 2012), and (v) oxidative C(sp3)-/Se coupling in tetralones (Prasad, Sattar & Kumar, 2017). Despite these are efficient methodologies, chlorinated or high boiling point solvents, harsh base, transition metal catalysts, and/or halogenating reagents are involved in the synthesis.

On the other hand, Oxone® is an inexpensive, stable, water-soluble, and safe alternative oxidizing agent that has been used in numerous oxidation reactions (Hussain, Green & Ahmed, 2013). This green oxidant is a mixture of three inorganic salts (2KHSO5·KHSO4·K2SO4), with potassium peroxymonosulfate (KHSO5) being the active species. The synthesis of important heterocyclic compounds was accomplished using Oxone®, such as chromene and carbazoles (Reddy, Kannaboina & Das, 2017), benzimidazoles (Daswani et al., 2016), benzoxazoles (Hati et al., 2016), pyrazole (Kashiwa et al., 2016), and pyridine derivatives (Swamy et al., 2016). Furthermore, it was used in intramolecular cycloaddition (More & Ramana, 2016) and cyclization reactions (Sharma et al., 2016), in the synthesis of α-bromoketones (Rammurthy et al., 2017), in halogenation reactions of quinolines (Wang et al., 2016), oxidation of alcohols to carbonyl compounds (Mishra & Moorthy, 2017) and in the synthesis of iodohydrins and iodoarenes (Soldatova et al., 2016). However, to the best of our knowledge, no reactions using Oxone® to prepare electrophilic selenium species as substrate in cycloaddition reactions have been described so far.

In the last years, the use of ultrasonic waves as an alternative energy source in organic synthesis has exponentially increased. The so-called sonochemistry has the ability to accelerate, or even totally modify the reaction course, through the formation of new reactive intermediates that normally are not involved when conventional heating is used (Nowak, 2010; Mojtahedi & Abaee, 2012; Schiel et al., 2015). Recently, we have described new ultrasonic-promoted reactions, including the synthesis of 1,2,3-triazoyl carboxamides (Xavier et al., 2017), 3-selanylindoles (Vieira et al., 2015) and chrysin derivatives (Fonseca et al., 2017). Considering the importance of organoselenium compounds and naphthalene derivatives, and due our interest in green synthetic protocols associated to organochalcogen chemistry, we report herein a new ultrasound-promoted method to prepare 2-organoselanyl-naphthalenes 3a–i. Our strategy involves the carbocyclization of alkynols 2a–d using electrophilic selenium species, which were generated in situ by the reaction of diorganyl diselenides 1a–f with Oxone® (Fig. 1).

Figure 1 Synthesis of 2-organochalcogenyl-naphtalenes.

Material and Methods

General remarks

Pre-coated TLC sheets (ALUGRAM® Xtra SIL G/UV254; Macherey-Nagel GmbH & Co-KG, Düren, Germany). using UV light and acidic ethanolic vanillin solution (5% in 10% H2SO4) were used to follow the reaction progress. Aldrich technical grade silica gel (pore size 60 Å, 230–400 mesh) was used for flash chromatography. Carbon-13 nuclear magnetic resonance (13C NMR) and hydrogen nuclear magnetic resonance spectra (1H NMR) were obtained on Bruker Ascend 400 spectrometers at 100 MHz at 400 MHz, respectively. Spectra were recorded in CDCl3 solutions. Chemical shifts are reported in ppm, referenced to tetramethylsilane (TMS) as the internal reference, for 1H NMR and the solvent peak of CDCl3 for 13C NMR. Coupling constant (J) are reported in hertz. Abbreviations to denote the multiplicity of a particular signal are brs (broad signal), s (singlet), d (doublet), dd (doublet of doublet), t (triplet), and m (multiplet). A Shimadzu GC-MS-QP2010 was used to obtain the low-resolution mass spectra (MS), while a LTQ Orbitrap Discovery mass spectrometer (Thermo Fisher Scientific, Waltham, MA, USA) was employed to obtain the high-resolution mass spectra (HRMS), the experiments were performed via direct infusion of sample (flow: 10 μL/min) in the positive-ion mode using electrospray ionization. A (Cole Parmer CPX 130; Cole-Parmer Instrument Company, Chicago, IL, USA) operating with an amplitude of 60%, maxim power of 130 W at 20 KHz, was used to generate the ultrasonic waves. The temperature of the reactions under US was monitored with a Incoterm digital infrared thermometer (Infraterm, São Paulo, Brazil). Melting point (m.p.) values were measured in a Marte PFD III instrument with a 0.1 °C precision. Oxone® was purchased from (Sigma-Aldrich, St. Louis, MO, USA).

General procedure for the synthesis of 2-organoselanyl-naphthalenes 3

To a 10 mL round bottomed glass tube, the appropriate diorganyl diselenide 1a–f (0.125 mmol), alkynol 2a–d (0.25 mmol), water (2.0 mL), and Oxone® (0.077 g; 0.25 mmol) were added. The US probe was placed in the reaction vial, which was sonicated (20 KHz, 60% of sonic amplitude) for the time indicated in Figs. 2 and 3. The reaction temperature was monitored and after 5 min it was around 64–65 °C, which was maintained until the end of the reaction. The reaction progress was monitored by TLC in order to evaluate the starting materials consumption. After the reaction was completed, the reaction mixture was extracted with ethyl acetate (15.0 mL), the organic phase was separated, dried over MgSO4, filtered and the solvent was evaporated under reduced pressure. The product was purified by column chromatography using hexanes as the eluent (except for 3f, where a mixture EtOAc/hexane (40/60) was used). All the compounds were properly characterized by MS, 1H NMR, 13C NMR, and HRMS (for the new ones).

Figure 2 Optimization of reaction conditions to prepared compound 3aa.

aA mixture of 1a (0.125 mmol), 2a (0.25 mmol), Oxone®, and the solvent (2.0 mL) in a glass tube was sonicated for the time indicated in the figure; the final temperature was 65 °C. bIsolated yields after column chromatography. cReaction performed under conventional heating (oil bath at 60 °C) under magnetic stirring. dIt was used 0.30 mmol of 2a. eKHSO4 (0.25 mmol) was added to the reaction mixture. NR, no reaction.

Figure 3 Synthesis of 2-organochalcogenyl-naphthalenes 3a–ia.

aThe mixture of reagents 1 (0.125 mmol), 2 (0.25 mmol), Oxone® (0.25 mmol) and 2.0 mL of water was added to the glass tube and sonicated for the time indicated in the figure. bYields of isolated products after column chromatography.

1-Phenyl-2-phenylselanyl-naphthalene (3a) (Recchi, Back & Zeni, 2017): yield: 0.077 g (86%); yellowish solid; m.p. = 100–101 °C. 1H NMR (CDCl3, 400 MHz) δ = 7.80-7.78 (m, 1H); 7.64 (d, J = 8.8 Hz, 1H); 7.54-7.23 (m, 14H). 13C NMR (100 MHz, CDCl3) δ = 139.7, 139.6, 135.0, 133.1, 132.2, 131.0, 130.5, 130.2, 129.4, 128.5, 128.1, 127.9, 127.86, 127.8, 126.4, 126.1, 125.5. MS: m/z (rel. int., %) 360 (92.4), 280 (66.2), 202 (100.0), 126 (2.8), 77 (7.6).

2-(4-Chlorophenylselanyl)-1-phenyl-naphthalene (3b) (Recchi, Back & Zeni, 2017): yield: 0.062 g (63%); yellowish solid; m.p. = 117–118 °C. 1H NMR (CDCl3, 400 MHz) δ = 7.81 (d, J = 8.1 Hz, 1H); 7.68 (d, J = 8.7 Hz, 1H); 7.54-7.24 (m, 13H). 13C NMR (CDCl3, 100 MHz) δ = 140.2, 139.5, 136.0, 134.2, 133.1, 132.3, 130.3, 130.1, 129.6, 128.9, 128.5, 128.3, 128.2, 127.9, 126.6, 126.2, 125.7. MS: m/z (rel. int., %) 394 (68.9), 314 (45.1), 202 (100.0), 126 (3.5), 77 (3.5).

2-(4-Fluorophenylselanyl)-1-phenyl-naphthalene (3c) (Recchi, Back & Zeni, 2017): yield: 0.067 g (71%); yellowish solid; m.p. = 123–124 °C. 1H NMR (CDCl3, 400 MHz) δ = 7.80-7.78 (m, 1H); 7.65 (d, J = 8.7 Hz, 1H); 7.55-7.48 (m, 5H); 7.44-7.40 (m, 2H); 7.36-7.34 (m, 3H); 7.18 (d, J = 8.2 Hz, 1H); 7.00 (t, J = 8.8 Hz, 2H). 13C NMR (CDCl3, 100 MHz) δ = 162.9 (d, J = 246.6 Hz), 139.4, 139.2, 137.5 (d, J = 7.9 Hz), 133.0, 132.0, 131.1, 130.1, 128.5, 128.2, 127.9, 127.87, 127.4, 126.5, 126.0, 125.5, 124.7 (d, J = 3.5 Hz), 116.7 (d, J = 21.2 Hz). MS: m/z (rel. int., %) 378 (74.2), 298 (65.2), 202 (100.0), 126 (2.3), 77 (1.9).

2-Mesitylselanyl-1-phenyl-naphthalene (3d) (Recchi, Back & Zeni, 2017): yield: 0.078 g (78%); yellowish solid; m.p. = 111–112 °C. 1H NMR (CDCl3, 400 MHz) δ = 7.74 (d, J = 8.1 Hz, 1H); 7.58-7.31 (m, 9H); 6.99 (s, 2H); 6.82 (d, J = 8.7 Hz, 1H); 2.37 (s, 6H); 2.31 (s, 3H). 13C NMR (100 MHz, CDCl3) δ = 143.8, 139.7, 139.0, 137.9, 133.2, 132.0, 131.7, 130.0, 128.9, 128.6, 128.0, 127.9, 127.8, 127.7, 126.3, 125.5, 125.2, 124.9, 24.2, 21.1. MS: m/z (rel. int., %) 402 (100.0), 202 (55.9), 198 (57.3), 91 (18.4), 77 (9.3).

1-Phenyl-2-(2-pyridylselanyl)-naphthalene (3e): yield: 0.076 g (84%); yellowish oil; 1H NMR (CDCl3, 400 MHz) δ = 8.42-8.40 (m, 1H); 7.87-7.85 (m, 1H); 7.79 (d, J = 8.6 Hz, 1H); 7.73 (d, J = 8.6 Hz, 1H); 7.50-7.35 (m, 7H); 7.29-7.27 (m, 2H); 7.10 (d, J = 8.0 Hz, 1H); 7.03-7.00 (m, 1H). 13C NMR (100 MHz, CDCl3) δ = 158.1, 150.0, 143.5, 139.9, 136.5, 133.3, 133.1, 131.9, 130.0, 128.5, 128.1, 127.9, 127.7, 127.6, 127.0, 126.4, 126.2, 126.0, 120.7. MS: m/z (rel. int., %) 361 (56.7), 284 (100.0), 278 (13.8), 202 (74.8), 79 (16.4). HRMS calcd. for C21H15NSe: [M+H]+ 362.0448; found: 362.0443.

1-Phenyl-2-(propanyl-2,3-diolselanyl)-naphthalene (3f): yield: 0.050 g (56%); yellowish oil; 1H NMR (CDCl3, 400 MHz) δ = 7.83 (d, J = 8.4 Hz, 1H); 7.79 (d, J = 8.6 Hz, 1H); 7.66 (d, J = 8.6 Hz, 1H); 7.54-7.28 (m, 8H); 3.73-3.63 (m, 3H); 3.48 (dd, J = 11.1 and 5.9 Hz, 1H); 3.01 (dd, J = 12.8 and 4.7 Hz, 1H); 2.89 (dd, J = 12.8 and 8.0 Hz, 1H); 2.59 (br, 1H). 13C NMR (100 MHz, CDCl3) δ = 141.6, 139.8, 133.0, 132.3, 130.3, 130.0, 128.5, 128.4, 128.37, 128.2, 127.9, 127.2, 126.6, 126.3, 125.8, 70.2, 65.5, 31.5. MS: m/z (rel. int., %) 358 (55.9), 280 (46.8), 202 (100.0). HRMS calcd. for C19H18O2Se: [M]+ 358.0472; found: 358.0467.

2-Phenylselanyl-1-(4-tolyl)-naphthalene (3g) (Recchi, Back & Zeni, 2017): yield: 0.059 g (63%); yellowish oil; 1H NMR (CDCl3, 400 MHz) δ = 7.78 (d, J = 8.0 Hz, 1H); 7.63 (d, J = 8.8 Hz, 1H); 7.53-7.51 (m, 2H); 7.46-7.39 (m, 2H); 7.36-7.23 (m, 9H); 2.47 (s, 3H). 13C NMR (CDCl3, 100 MHz) δ = 139.5, 137.5, 136.5, 135.1, 133.2, 132.1, 131.1, 130.4, 130.0, 129.4, 129.2, 128.0, 127.9, 127.89, 127.8, 126.4, 126.1, 125.4, 21.4. MS: m/z (rel. int., %) 374 (100.0), 282 (18.5), 202 (52.3), 91 (2.0).

1-(4-Chlorophenyl)-2-phenylselanyl-naphthalene (3h) (Recchi, Back & Zeni, 2017): yield: 0.071 g (72%); yellowish oil; 1H NMR (CDCl3, 400 MHz) δ = 7.80 (d, J = 8.1 Hz, 1H); 7.66 (d, J = 8.7 Hz, 1H); 7.49-7.27 (m, 13H). 13C NMR (CDCl3, 100 MHz) δ = 138.6, 138.0, 134.9, 133.9, 133.0, 132.2, 131.6, 131.0, 130.3, 129.4, 128.7, 128.5, 128.4, 128.0, 127.99, 126.7, 125.8, 125.7. MS: m/z (rel. int., %) 394 (100.0), 282 (25.4), 202 (69.6), 126 (2.5), 77 (5.4).

2-Phenylselanyl-1-propyl-naphthalene (3i): yield: 0.094 g (94%); yellowish oil; 1H NMR (CDCl3, 400 MHz) δ = 8.05 (d, J = 8.7 Hz, 1H); 7.79-7.77 (m, 1H); 7.54-7.42 (m, 5H); 7.27-7.25 (m, 3H); 3.34-3.30 (m, 2H); 1.76-1.66 (m, 2H); 1.09 (t, J = 7.3 Hz, 3H). 13C NMR (CDCl3, 100 MHz) δ = 140.6, 133.2, 132.8, 132.3, 131.8, 131.5, 131.4, 129.31, 129.26, 129.2, 127.7, 127.1, 127.0, 126.4, 125.6, 124.4, 34.6, 24.1, 14.5. MS: m/z (rel. int., %) 326 (61.5), 216 (100.0), 202 (10.9), 77 (3.4). HRMS calcd. for C19H18Se: [M+H2O+H]+ 345.0758; found: 345.0753.

Results and Discussion

The selenocyclization of alkynols with electrophilic selenium species is an efficient strategy to prepare organoselanyl-naphthalenes (Recchi, Back & Zeni, 2017). In our preliminary studies on the use of Oxone® as an oxidant to cleavage of Se–Se bond, we have observed that its reaction with diselenides generates highly reactive species in situ (Perin et al., 2018). Thus, by combining the selenocyclization strategy with the environmental and economic advantages of using Oxone® as an oxidizing agent, a study was carried out to evaluate the possibility of using it in selenocyclization reactions to prepare organoselanyl-naphthalenes. In our preliminary experiments, we choose diphenyl diselenide 1a and 1,4-diphenylbut-3-in-2-ol 2a as model substrates to establish the best conditions for the cyclization reaction promoted by Oxone® to synthesize the respective 2-organoselanyl-naphthalene 3a.

Initially, the reaction was performed using 0.25 mmol of alkynol 2a, 0.125 mmol of diphenyl diselenide 1a and 0.25 mmol of Oxone®, using ethanol (2.0 mL) as the solvent at 60 °C under magnetic stirring. The desired product 3a was obtained in 78% yield after 72 h (Fig. 2, entry 1). To improve this result, some experiments were performed with the purpose of increasing the isolated yield and reducing the reaction time. The same reaction was then performed under ultrasonic irradiation (amplitude of 60%) and after 50 min, product 3a was obtained in 84% yield (Fig. 2, entry 2). Aiming to improve the yield of 3a, parameters as the nature of the solvent, quantities of the starting material 2a, amounts of Oxone®, and amplitude of the US were evaluated (Fig. 2, entries 3–12).

Regarding the influence of the solvent in the reaction, a range of solvents were tested and in reactions using polyethylene glycol-400 (PEG-400, Labsynth, Diadema, Brazil), glycerol, and DMF, product 3a was obtained in good yields (Fig. 2, entries 3, 4, and 8). To our satisfaction, a very good yield of 86% was obtained after sonication of the reaction mixture for 30 min in water (Fig. 2, entry 5). However, using dimethyl sulfoxide (DMSO) or acetonitrile as the solvent, only trace amounts of 3a were observed (Fig. 2, entries 6 and 7).

After water was defined as the best solvent for the reaction, the amplitude used in the ultrasound apparatus was evaluated. When the reaction was performed at 40% of amplitude, the desired product 3a was obtained in only 63% yield (Fig. 2, entry 9). It was observed that at this lower amplitude, the homogenization of the mixture was incomplete, what could negatively affect the reaction yield.

When an excess of alkynol 2a was used, total consumption of diphenyl diselenide 1a occurred after 30 min of reaction (monitored by TLC), however the yield of 3a was maintained (Fig. 2, entry 10). By using a lower amount of Oxone® (0.125 mmol), there was no total consumption of the starting materials after 2 h of reaction, and the desired product 3a was obtained in only 42% yield (Fig. 2, entry 11). Finally, the reaction was carried out in the absence of Oxone® and after 2 h none of product was formed (Fig. 2, entry 12). In order to verify the influence of the KHSO4 species present in the reaction medium, a test was performed using 0.25 mmol of Oxone® together with 0.25 mmol of KHSO4 and, after only 10 min of reaction, the starting materials were totally consumed, and the desired product 3a was obtained in 92% isolated yield, showing the need of generation of this species in the reaction medium (Fig. 2, entry 13). Thus, the best condition was defined as the sonication of a mixture of 0.125 mmol of diphenyl diselenide 1a and 0.25 mmol of alkynol 2a in the presence of 0.25 mmol of Oxone® in water (2.0 mL) for 30 min (Fig. 2, entry 5).

Once the best reaction conditions were determined, the methodology was extended to different substrates, in order to evaluate its generality and robustness in the synthesis of different 2-organoselanyl-naphthalenes 3a–i (Fig. 3). Firstly, the effect of electron-donor (EDG) and electron-withdrawing groups (EWG) attached to the aromatic ring of diselenide 1a–d was evaluated (Fig. 3, entries 1–4). It was observed that both EDG and EWG negatively affect the reaction, affording lower yields of the respective products. When diselenide 1b, containing a chlorine atom at the para position was used, there was a significant decrease in yield when compared to diphenyl diselenide 1a, and the respective naphthalene 3b was obtained in 63% yield (Fig. 3, entry 2). Similarly, the electron-poor diselenide 1c, with a fluorine atom at the para position, afforded the respective naphthalene 3c in a moderate yield of 71% after 1.4 h (Fig. 3, entry 3).

The sterically hindered dimesityl diselenide 1d was also a suitable substrate for the reaction, affording the expected product 3d in 78% yield after 0.5 h of sonication (Fig. 3, entry 4). Heteroaromatic bis-pyridyl diselenide 1e was successfully used as substrate in the reaction with alkynol 2a, affording the respective 2-heteroarylselanyl-naphthalene 3e in 84% yield (Fig. 3, entry 5).

Interestingly, when diselenide derived from protected glycerol (solketal) 1f was used, deprotected naphthalene diol 3f was obtained in 56% yield after 0.5 h of reaction (Fig. 3, entry 6). This may be associated with the ketal deprotection ability of Oxone®, which has already been reported in the literature (Mohammadpoor-Baltork, Amini & Farshidipoor, 2000).

The possibility of performing these reactions with other alkynols 2b–d was also investigated. Alkynols derived from phenylacetylene 2b and 2c, containing EDG and EWG at the aromatic ring, efficiently reacted with diphenyl diselenide 1a/Oxone®, affording the respective products 3g and 3h in 63 and 72% yields after 1.7 and 2.3 h, respectively (Fig. 3, entries 7 and 8). This result shows that the reaction is not sensitive the electronic effects of the substituents on the aromatic ring of the alkynols 2b and 2c. A remarkable positive effect was observed when an alkyl group was connected to the Csp of the alkynol, as in 2d and an excellent 94% yield of the expected naphthalene 3i was obtained after 1.0 h (Fig. 3, entry 9).

Based on our results and those from the literature (Zhang, Sarkar & Larock, 2006; Recchi, Back & Zeni, 2017; Perin et al., 2018), a plausible mechanism for the carbocyclization of alkynol 1a with (C6H5Se)2 2a/Oxone® in aqueous medium is depicted in Fig. 4. The first step in the reaction is the oxidative cleavage of the Se–Se bond in diphenyl diselenide 2a by Oxone®, forming intermediates A and B (Perin et al., 2018). Once the electrophilic selenium species A is formed, it reacts with the carbon–carbon triple bond of the alkynol 1a to produce the seleniranium intermediate C. Following, an intramolecular 6-endo-dig cyclization occurs, giving intermediate D, which undergoes deprotonation to restoring the aromaticity of the system, forming the dihydronaphthalene E. Ultimately, water is eliminated to give the desired product 3a (Fig. 4).

Figure 4 Proposed mechanism.

Conclusion

A convenient, selective and eco-friendly methodology was developed for the synthesis of 2-organoselanyl-naphthalenes 3, using water as the solvent. The use of ultrasound as alternative energy source drastically reduces the reaction time, while increasing the reaction yield. This method involves the cyclization of properly substituted alkynols in the presence of electrophilic selenium species. Oxone® was shown to be an efficient and mild oxidizing agent for the oxidative cleavage of the Se–Se bond of diselenides in situ.

Supplemental Information

Supplemental Information 1 Supplemental material–Spectra data and figures for synthesized compounds (raw data).

Click here for additional data file.

Additional Information and Declarations

Competing Interests

Author Contributions

Data Availability

Eder J Lenardao is an Academic Editor for PeerJ.

Gelson Perin conceived and designed the experiments, analyzed the data, contributed reagents/materials/analysis tools, authored or reviewed drafts of the paper, approved the final draft.

Daniela Rodrigues Araujo performed the experiments, prepared figures and/or tables.

Patrick Carvalho Nobre performed the experiments, prepared figures and/or tables.

Eder João Lenardao conceived and designed the experiments, analyzed the data, contributed reagents/materials/analysis tools, authored or reviewed drafts of the paper, approved the final draft.

Raquel Guimarães Jacob analyzed the data, contributed reagents/materials/analysis tools, authored or reviewed drafts of the paper.

Marcio Santos Silva performed the experiments, analyzed the data, contributed reagents/materials/analysis tools, authored or reviewed drafts of the paper.

Juliano Alex Roehrs performed the experiments, prepared figures and/or tables.

The following information was supplied regarding data availability:

The raw data are provided in the Supplemental File.

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
