# Peer review of "Ultrasound-promoted synthesis of 2-organoselanyl-naphthalenes using Oxone® in aqueous medium as an oxidizing agent"

_PeerJ, doi:10.7717/peerj.4706_

## Round 0.1 · original submission · Minor Revisions

Please address the reviewers comments

·

Basic reporting

The manuscript is written in a good level of language which makes it clear and understandable.
The introduction section is informative with a reasonable length. The literature is well explained with relevant reference; however, more references are required to support some important information (see the general comments below).
In general, the structure of the manuscript fits to PeerJ standards (considering the suggested corrections and explanations).
The figures are relevant and informative but the need to be uniformed and with a better quality!
The appropriate raw data are available.

Experimental design

The work is original within the area of green methodologies of organic synthesis. The manuscript describes the development of a new methodology for the synthesis of an important class of organic compounds (2-organoselanyl-naphthalenes) using water as solvent as green solvent with a relatively short reaction times!
The experiments are well designed in a good sequence of work steps that showed how to get the optimum conditions of the developed methodology followed by proving the applicability of the method on deferent substrates which were chosen to cover deferent structural properties. The new method is well described with sufficient details, the variety of the substrates and obtained results indicate that the method is reproducible!

Validity of the findings

Positive results are reported, proving the validity of the newly developed method.
Clear raw data are supplied. The choices of the different substrates cover a broad spectrum of molecular structures properties!
The conclusions are well stated and connected to the core of the work and the obtained results.

Additional comments

Lines 46-47: “The reaction of diorganyl diselenides with a halogen source is the most used method to access electrophilic selenium species.”
References are required to support this statement.

Line 49: “alternative methodologies”
Explain some of these methodologies supported by recent refernces.

Lines 48-52: “This point has motivated the development of alternative methodologies to synthesize electrophilic species of selenium and a greener alternative to selanyl halides are the convenient sulfate analogues, formed in situ by the reaction of diaryl diselenides with ammonium (Tiecco et al., 1989) or potassium persulfate (Santi & Tidei, 2014).”
The statement need to be re-written to make it clearer to the reader!

Line: 109: NMR Spectrometer brand to be mentioned.

Line: 110: are the chemical shift values “referenced to the solvent peak of CDCl3” or to TMS?!

Line 115: “(HRMS). The experiments” to be replaced by “(HRMS), the experiments”

Line 130: “dried with MgSO4” to be replaced by “dried over MgSO4”

Line 132: “hexanes” to be replaced by “hexane”

Line 136: [33] should be referring to the reference (Recchi, Back & Zeni, 2017), follow the author guidelines consistently, the same mistake in Lines 143, 150 and 158.

Line 144: The spectrum in the raw data supplied shows that the signal at 7.82-7.80 d rather than m, it could be reported app. d!
The same for 7.73 (Line 159), 7.79-7.77 (Line 181) and 8.07-8.05 ; 7.79-7.77 (Line 194).

Lines 153-154: There is a mismatch with the provided spectrum in the raw data!

Line 212: Is it 0.25 or 0.125 mmol of Oxone?! (Check table 1).

Lines 217-218:” Aiming to improve the yield of 3a, different amounts of starting material 2a
218 and Oxone®, as well as the nature of the solvent, were evaluated (Table 1, entries 3-12).”
Table 1 shows that 0.125 mmol of 2a was used in all cases!
Referring to entries 3-12 is not accurate, because in entry 9 the US amplitude was changed!

Lines 236-237: 0.125 mmol of Oxone; Line 241: 0.25 mmol of Oxone; there is a contradiction even with Table 1 (check entries 1 and 13)!.

Line 249: 3 to be replaced by 3a-i

Line 249: 1 to be replaced by 1a-d

Lines 269-270: “it was not possible to infer any electronic effect of substituents in the aromatic ring of alkynols 2b and 2c.”
The statement is not clear!

Line 271: Csp, what does it mean?
Line 272: “entry 11” to be replaced by “entry 9”.
Other Points and Questions:
Correct the caption of Table 1.
For the reactions with relatively moderate yields obtained, have the authors tried to improve the yields by giving the reactions longer time to proceed?
In some of the 1H NMR spectra (compounds 3b-d) signal at around 5.1 ppm is observed, any explanation?

In conclusion the manuscript can be considered for publication in PeerJ after minor corrections (considering the points mentioned above).

Reviewer 2 ·

Basic reporting

The article is written in good and clear English level.
The article fits the PeerJ standards with some minor mistakes as shown in the general comments for the author.

Experimental design

The experiments are well designed with good reaction sequence steps.

Validity of the findings

No comment

Additional comments

Need to double check the matching of the manuscript data and table 1 for example:
Line 212: not matching with Entry 1 (Table 1).

Some minor mistakes in references formatting, for example:
Line 344: Year of Publication (not written).
Line: 363: Name of the book (shouldn't be italic).
Line 374: Name of the journal (should be italic).
Line 375: space.
Line 382: No italic.
Lines 390, 422 and 423.

The manuscript may very well be suitable for publication in PeerJ after making sure of previous observations and correcting some minor errors.

·

Basic reporting

The manuscript is very well written with nice finding and I recommend it for publications

Experimental design

no comment

Validity of the findings

no comment

Additional comments

• Abstract: is seems very well written
• Introduction: it is seems that the introduction cover the most important papers and the recent one but it is too long, I suggest to reduce it
• In line 76: you should use “comma” and “and” (2KHSO5, KHSO4 and K2SO4) instead of (2KHSO5.KHSO4.K2SO4) because it is mixture as you said in the same line.
• Material and method: clear with small comment:
• Line 108: it is better to replace hydrogen by proton
• Line 109: you need to write the instruments brand and model
• Results and discussion: no comments
• Conclusion: no comments
• References: no comments
Overall, the manuscript is very well written with nice finding and I recommend it for publications

---

## Round 0.2 · accepted · Accept

Dear Professor Eder
Thank you for your submission to PeerJ.

I am writing to inform you that your manuscript - Ultrasound-promoted synthesis of 2-organoselanyl-naphthalenes using Oxone® in aqueous medium as oxidizing agent - has been Accepted for publication.

Best regards
Gamal El-Hiti

#